
# Enhancing sensitivity to rotations with quantum solitonic currents

P. Naldesi[1*], J. Polo[1,2,3], V. Dunjko[4], H. Perrin[5],
M. Olshanii[4], L. Amico[3,6,7,8,9,10] and A. Minguzzi[1]

**1** Univ. Grenoble-Alpes, CNRS, LPMMC, 38000 Grenoble, France
**2** Quantum Systems Unit, Okinawa Institute of Science and Technology Graduate University, Onna, Okinawa 904-0495, Japan
**3** Quantum Research Centre, Technology Innovation Institute, Abu Dhabi, UAE
**4** Department of Physics, University of Massachusetts Boston, Boston, MA 02125, USA
**5** Laboratoire de physique des lasers, CNRS UMR 7538 and Université Sorbonne Paris Nord, av. J.-B. Clément, F-93430 Villetaneuse, France
**6** Dipartimento di Fisica e Astronomia, Via S. Sofia 64, 95127 Catania, Italy
**7** Centre for Quantum Technologies, National University of Singapore, 3 Science Drive 2, Singapore 117543, Singapore
**8** MajuLab, CNRS-UNS-NUS-NTU International Joint Research Unit, UMI 3654, Singapore
**9** CNR-MATIS-IMM & INFN-Sezione di Catania, Via S. Sofia 64, 95127 Catania, Italy
**10** LANEF *'Chaire d'excellence'*, Université Grenoble-Alpes & CNRS, F-38000 Grenoble, France

⋆ piero.Naldesi@uibk.ac.at

## Abstract

We study a gas of attracting bosons confined in a ring shape potential pierced by an artificial magnetic field. Because of attractive interactions, quantum analogs of bright solitons are formed. As a genuine quantum-many-body feature, we demonstrate that angular momentum fractionalization occurs and that such an effect manifests on time of flight measurements. As a consequence, the matter-wave current in our system can react to very small changes of rotation or other artificial gauge fields. We worked out a protocol to entangle such quantum solitonic currents, allowing us to operate rotation sensors and gyroscopes to Heisenberg-limited sensitivity. Therefore, we demonstrate that the specific coherence and entanglement properties of the system can induce an enhancement of sensitivity to an external rotation.

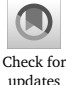

# 1  Introduction

Quantum coherence and quantum correlations in many-body systems are fundamental features that separate quantum systems from classical ones. These features have been of central importance in the development of quantum optics [1], mesoscopic physics [2], and quantum material science [3]; and they are now at the heart of quantum technology. Indeed, the defining goal of quantum technology is to realize new concepts of quantum devices and simulators harnessing quantum coherence and entanglement [4–6].

A natural way to access the resources needed for quantum technologies is to refer to quantum many-body systems in the strongly correlated regime. For example, superconducting circuits and circuit QED rely on the quantum coherence resulting from the specific electronic (pairing) correlations occurring in superconductors [7]. In precision measurement, many-body correlations have recently been used in optical lattice clocks to prepare isolated atoms [8], allowing in turn to measure many-body effects with clock precision [9]. With atomic ensembles, massive particle entanglement has enabled a noise reduction by a factor of 100 in a microwave clock system [10]. Our system is made of attracting neutral bosonic atoms flowing in a ring-shaped lattice potential of mesoscopic size, which sustains a neutral persistent current flow (see Fig. 1). As a physical implementation of such a system, we propose ultra-cold atoms [11], with a new twist provided by atomtronics [12, 13].

In contrast with continuous systems, lattice rings provide a characteristic energy-band structure, displaying bendings, foldings and energy gaps. Such features lead to a specific protection of the bright solitons [14]. On the other hand, we shall see that the lattice system provides a nontrivial generalization of a theorem due to Leggett [15] that predicts the characteristic response to an applied (artificial) magnetic field in quantum rings.

While our discussions apply to any type of artificial gauge fields [16], in the following we will refer to the case of an artificial gauge field induced by a global rotation at angular

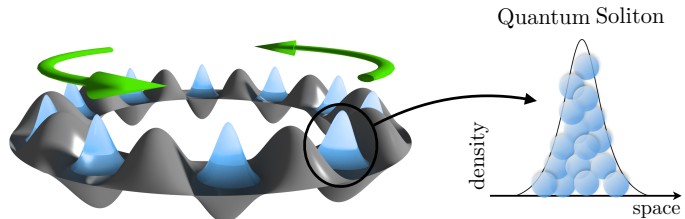

Figure 1: Schematic representation of the system. Left: Ring lattice of bosons with attractive interactions subjected to an artificial gauge field inducing matter-wave currents (arrows). Attractive interactions give rise to the formation of many-body bound states, i.e. quantum analogs of bright solitons, where many particles are clustered together (right).

frequency $\Omega$. For such systems, it is found that the induced angular momentum increases in quantized steps as a function of $\Omega$ [17, 18]; correspondingly, the amplitude of persistent currents displays periodic oscillations with $\Omega$ [19, 20], with a periodicity that Leggett proved to be fixed by the effective flux quantum of the system, irrespective of particle-particle interactions [15].

Below, we demonstrate that for strongly correlated one-dimensional bosons with attractive interactions, the very nature of flux quantum is nontrivial, due to the formation of many-body bound states. This feature has dramatic effects on the persistent current that oscillates with a periodicity $N$ times smaller than in the standard case corresponding to repulsive interactions. Remarkably, the periodicity depends on interaction, which leads to an extension of the Leggett theorem. We show how our system can be harnessed to construct specific entangled states of persistent currents characterised by sensitivity to the effective magnetic field reaching the Heisenberg limit (quantum advantage).

## 2 Physical model

### 2.1 Lieb-Liniger

Before treating the general case of the lattice ring, we will first assume that the density $D = N/L$ of bosons, where $N$ is the particle number and $L = 2\pi R$ is the perimeter of the ring of radius $R$, is small enough to describe the system through the continuous Bose-gas integrable theory or equivalently the Lieb-Liniger model [21]. For the lattice, that means small filling fractions $\nu = N/N_s = D\Delta$, with $\Delta = L/N_s$ being the lattice spacing. For such systems, we can apply exact results [22].

In the frame rotating at frequency $\Omega$, the Lieb-Liniger Hamiltonian reads

$$\hat{\mathcal{H}}_{LL} = \sum_{j=1}^{N} \frac{1}{2m} p_j^2 - \Omega L_z + g \sum_{j<l} \delta(x_j - x_l), \tag{1}$$

where $x_i$, $m$ and the $p_i$ are, respectively, the position, mass and the momentum of each particle, $L_z = \sum_{j=1}^{N} L_{z,j}$ is the total angular momentum of the $N$ particles and $g$ is the interaction strength. The Lieb-Liniger Hamiltonian can be recast to

$$\hat{\mathcal{H}}_{LL} = \sum_{j=1}^{N} \frac{1}{2m} \left( p_j - m\Omega R \right)^2 + g \sum_{j<l} \delta(x_j - x_l) + E_\Omega, \tag{2}$$

with a constant $E_\Omega = -Nm\Omega^2 R^2/2$. Here, we assume periodic boundary conditions.

Using a transformation to Jacobi coordinates (see Appendix) $\xi_l$ and their canonically conjugate momenta $Q_l$, where $\xi_N = X_{CM}$ and $Q_N = P_{CM}$ where $X_{CM} = (1/N)\sum_j x_j$ and $P_{CM} = \sum_j p_j$ are, respectively, the coordinate and momentum of the center of mass, we find that in the Hamiltonian (2) only the center-of-mass momentum is coupled to the artificial gauge field $\Omega$. Correspondingly, the many-body wavefunction can be written as $\Psi(x_1,...,x_N) = e^{i(P_{CM}-Nm\Omega R)X_{CM}/\hbar}\chi_{\text{relative}}(\xi_1,...,\xi_{N-1})$. In this case, $P_{CM} = \ell\hbar/R$ can take any value allowed by quantization of momentum in the ring ($\ell$ being an integer).

The ground-state energy reads $E_{GS} = \frac{1}{2Nm}(P_{CM}-Nm\Omega R)^2 + E_{int}$, where $E_{int}$ is the interaction energy of the fluid, which does not depend on $\Omega$.

For any repulsive interactions, $E_{GS}$ is periodic in $\Omega$ with a period $\Omega_0 = \hbar/mR^2$ (see Appendix). Therefore, the persistent current in the rotating frame $I_p = -(\Omega_0/\hbar)\partial E_{GS}/\partial\Omega$ reflects the center-of-mass quantization, and displays the characteristic sawtooth behaviour versus $\Omega$ [15], corresponding to a staircase behaviour of angular momentum $L_z$.

For attractive interactions the ground state is a many-body bound state, *i.e.* a 'molecule' made of $N$ bosons, corresponding to the quantum analog of a bright soliton [14, 23, 24]. This picture arises from the exact Bethe ansatz solution; within the regime of validity of the string hypothesis [24] (see Appendix) the ground state energy for arbitrary $\Omega$ reads

$$E_{GS} = \frac{\hbar^2}{2MR^2}\left(\ell - N\frac{\Omega}{\Omega_0}\right)^2 - \frac{N(N^2-1)g^2}{12}, \tag{3}$$

where the second term accounts for the interaction energy $E_{int}$. Therefore, the above equation shows that attracting bosons behave as a single massive object of mass $M = Nm$ under the effect of the artificial gauge field. The energy displays a $1/N$-periodicity as a function of the artificial gauge field, $\Omega$, in units of $\Omega_0$ corresponding to *fractionalisation* of angular momentum per particle. Transforming back to the non moving frame, we can observe that the average persistent current per particle, $\mathcal{I}_p$ is also reduced by a factor $N$

$$\mathcal{I}_p = \hbar\frac{\ell}{N}. \tag{4}$$

In analogy to the fractional quantum Hall effect, in our system, the elementary particles carrying a fraction of quantum of angular momentum are parts of composite objects. We shall see, however, that our composite object displays a very specific dependence on the interplay between interaction and system size.

## 2.2 Bose-Hubbard

We now discuss the general non-integrable case in which the lattice effects are relevant. We assume that the boson dynamics is entailed by the Bose-Hubbard Model (BHM) [25]:

$$\hat{\mathcal{H}}_{BH} = -J\sum_{j=1}^{N_s}\left(e^{-i\tilde{\Omega}}a_j^\dagger a_{j+1} + \text{h.c.}\right) + \frac{U}{2}\sum_{j=1}^{N_s}n_j\left(n_j-1\right), \tag{5}$$

where $a_j$ and $a_j^\dagger$ are site $j$ annihilation and creation Bose operators and $n_j = a_j^\dagger a_j$. The parameters $J$, $U$ in (5) are respectively the hopping amplitude and the strength of the on-site interaction, $N_s$ being the number of sites in the ring lattice and $\tilde{\Omega} \doteq 2\pi\Omega/(\Omega_0 N_s)$ for brevity. In this study we will restrain our focus to attractive interactions $U < 0$.

We point out that, for the lattice model (5), the center-of-mass and relative coordinates do not decouple (for any finite interaction). As an effect, the internal structure of the many-body bound state is affected by the interplay between interaction and artificial gauge field $\Omega$ (since $P_{CM}$ depends on $\Omega$, and the internal structure depends on $P_{CM}$).

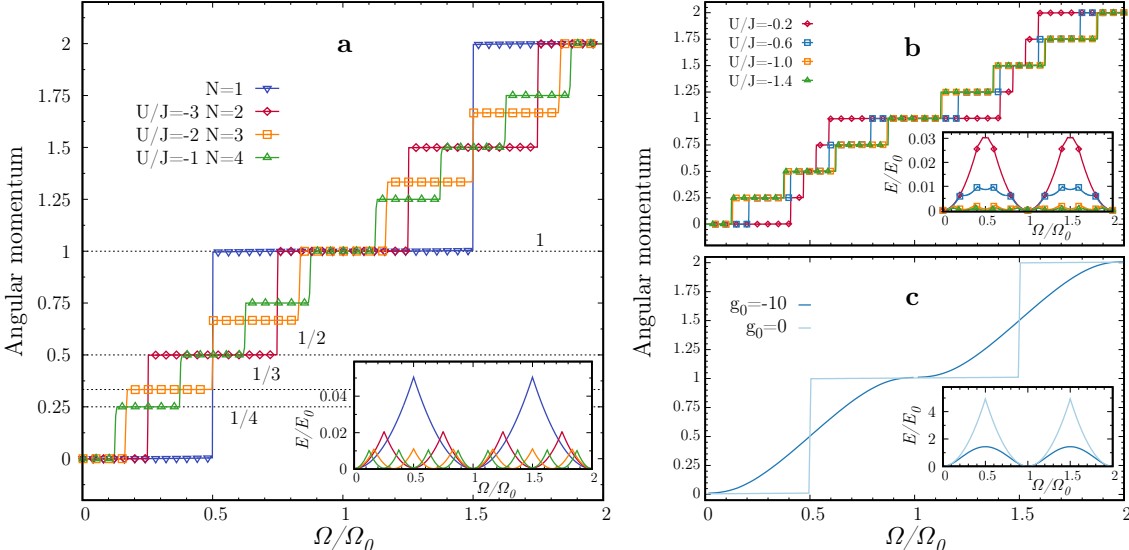

Figure 2: Average angular momentum per particle (main) and ground-state energy (inset) for bosons on a lattice ring as a function of artificial gauge field, from numerical exact diagonalization calculations: **a**) at varying particle number, for chosen values of interaction strength as indicated on the figure, **b**) for various values of interaction strength at fixed $N = 4$. Panel **c**) shows the corresponding predictions from the mean-field Gross-Pitaevskii equation for zero and finite attractive interactions indicated by the dimensionless parameter $g_0 = mgL/\hbar^2$. The angular momentum per particle is obtained as $\hbar\ell = \frac{M_{eff}}{M}\left(\frac{\hbar I_p}{\Omega_0} + \Omega\left.\frac{\partial^2 E_{GS}}{\partial\Omega^2}\right|_{\Omega=0}\right)$, with $M_{eff}$ being the effective mass of the bound state in the lattice. Simulation have been carried out on a system of $N_s = 14$ sites.

# 3 Results

## 3.1 Fractionalization of currents in the Bose–Hubbard model

Here, we find that the periodicity of the persistent current for lattice rings does depend on interaction. We remark that such a 'non-perfect' fractionalization (see Fig. 2(b)) is observed for solitons that are properly formed in the system (i.e. when the system size is larger than the correlation length of the density-density correlations). Fig. 2 shows our numerical results (confirmed by exactly solving the BHM in the 2-particle sector–see Appendix) for the ground-state energy, persistent currents and angular momentum: also in the lattice nonintegrable case the $1/N$ periodicity in $\Omega/\Omega_0$ of the persistent currents emerges, as well as fractionalization of angular momentum. Indeed, these features, though, are affected by the interplay between system size and interaction strength. The $1/N$ periodicity is found when interactions are sufficiently large: In these conditions, the 'size of the many-body bound state', defined as the typical decay length of the density-density correlations [14], is smaller than the size of the system. Upon decreasing the interactions, the many-body bound state spreads more and more over the sites making the solitonic nature of the state less and less pronounced (see Appendix). We remark that all the observed features are *purely quantum many-body effects tracing back to specific quantum correlations*: Indeed, mean-field Gross-Pitaevskii equation (corresponding to a non-entangled ground state) provides persistent currents displaying no fractionalization, independently on the strength of the interaction (see Fig. 2, c and Appendix F).

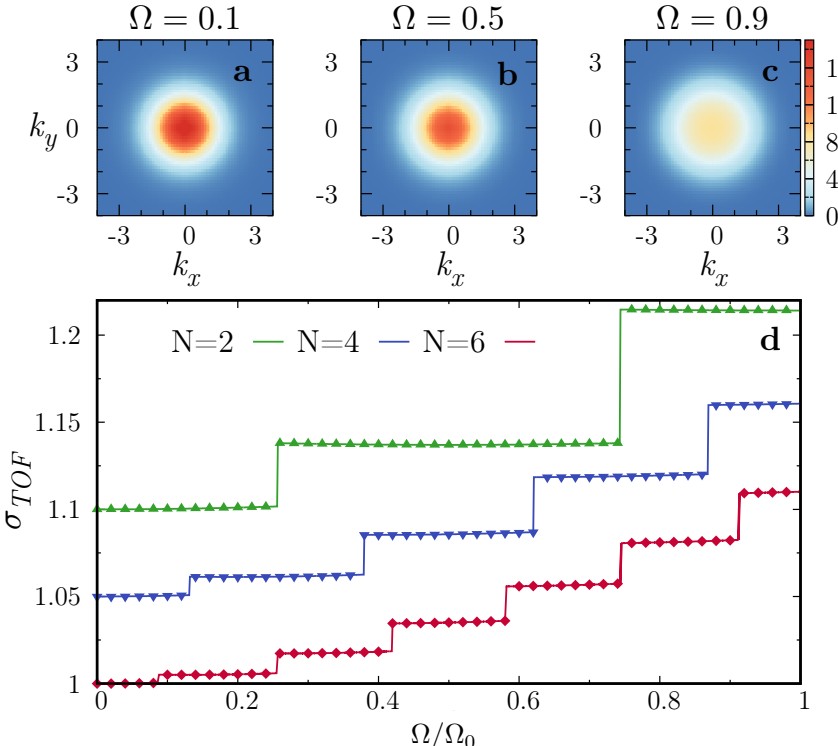

Figure 3: **a-b-c** Density plot of the TOF expansion for different values of the artificial gauge field in a system of $N = 4$ particles and $N_s = 11$ lattice sites. **d** Renormalized width $\sigma_{\text{TOF}}/\sigma_{\text{TOF}}(\Omega = 0)$ of the time of flight density distribution, $n(\mathbf{k})$. Green data are for $N = 2$ and $U/J = -3$, blue for $N = 4$ and $U/J = -1$ and red for $N = 6$ and $U/J = -0.6$. For the sake of graphical clarity, each curve is offset by 0.05. Note how the TOF density distribution width abruptly changes with the increase of the strength of the artificial gauge field, and how the sensitivity proportionally increases with the number of particles. In all the calculations we have approximated the Wannier functions with Gaussians functions with width $\delta = a/\sqrt{2\pi}$ with $a$ the lattice spacing.

## 3.2 Time of flight measurement

Remarkably, the afore discussed angular momentum fractionalization and persistent current periodicity emerge in the time-of-flight (TOF) distributions of the atoms after releasing the trap confinement and switching off interactions. We obtain it from

$$n(\mathbf{k}) = |w(\mathbf{k})|^2 \sum_{j,l} e^{i\mathbf{k}\cdot(\mathbf{x}_j - \mathbf{x}_l)} \langle a_j^\dagger a_l \rangle, \tag{6}$$

where $\mathbf{x}_j$ indicates the position of the lattice sites in the plane of the ring and $w(\mathbf{k})$ is the Fourier transform of the Wannier function of the lattice [26]. Instead of the characteristic wide $\ell$-dependent minimum ('hole') arising for zero or repulsive interactions [17,18], we find no clear hole at $\mathbf{k} = 0$ for the attractive case Fig. 3. Such a feature is due to the reduction of coherence implied by the solitonic many-body bound state. Despite the seemingly featureless momentum distribution, we find that *fractional steps of the mean-square radius of the distribution for $\Omega/\Omega_0 = \ell/N$ [27]. This effect provides the univocal signature of $1/N$ fractionalization of angular momentum in the presence of a many-body bound state.*

### 3.3 Entangling states with different angular momentum

We finally show how the scenario above can be harnessed to construct entangled states of different current states with quantum advantage for atom interferometry. For this purpose we follow the recipe proposed in [28] and illustrated in figure 4.

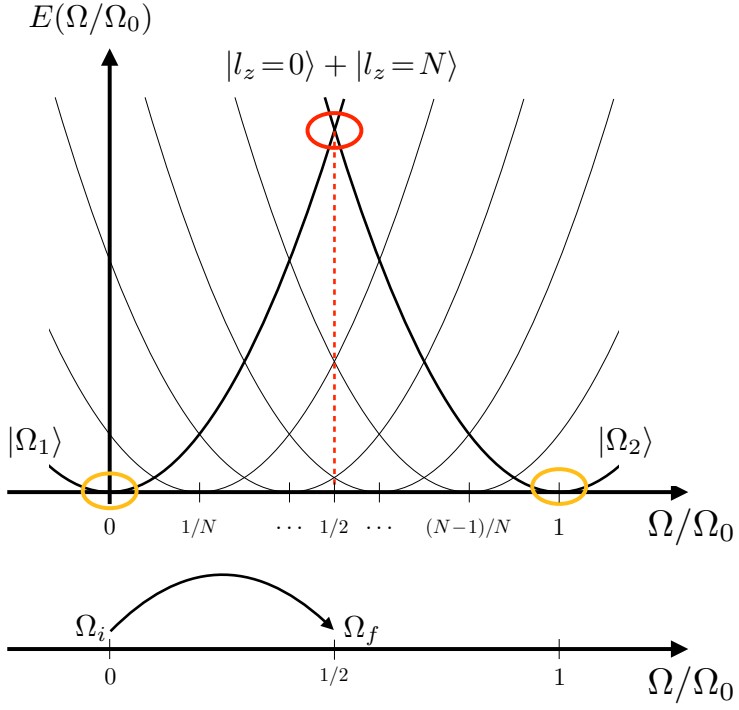

Figure 4: Schematic representation of the protocol. By quenching the flux in the system from 0 to $\Omega_0/2$, the two states $|\Omega_1\rangle$ and $|\Omega_2\rangle$ are dynamically entangled.

The protocol, described in the following, is based on the idea that by quenching the flux in the system, we can dynamically entangle any pair of ground states of the Hamiltonian (5) at different $\Omega/\Omega_0$. More precisely, let $|\Omega_1\rangle$ and $|\Omega_2\rangle$ be two ground states of (5) for $\Omega_1 = 0$ and $\Omega_2 = \Omega_0$ respectively. After initializing the system in $|\Omega_1\rangle$, the flux is quenched to $\Omega_f = (\Omega_2 - \Omega_1)/2 = \Omega_0/2$, exactly the value of flux for which such two states are degenerate in energy. Since, Hamiltonian Eq.(5) commutes with the total angular momentum, in order to entangle states with different angular momentum, the discrete rotational invariance of the system needs to be broken. The ring is then interrupted by adding a localized potential barrier of strength $\Delta_0$.

The state of the system will then evolve as:

$$|\psi(t)\rangle = e^{-i\hat{\mathcal{H}}_{BH}(\Omega_f)t} |\Omega_1\rangle. \tag{7}$$

To analyse the nature of the state during the time evolution, we calculate its fidelity over the equal superposisiton of the two target states, $|\Omega_1\rangle$ and $|\Omega_2\rangle$, as well as the current in the ring.

$$\mathcal{F}(t) = |\langle\psi(t)|\psi_{NOON}\rangle|^2 \quad \text{with} \quad |\psi_{NOON}\rangle = \frac{1}{\sqrt{2}}(|\Omega_1\rangle + |\Omega_2\rangle). \tag{8}$$

$$I(t) = -iJe^{-i\tilde{\Omega}_1} \sum_j \langle\psi(t)|a^\dagger_{j+1}a_j - \text{h.c.}|\psi(t)\rangle. \tag{9}$$

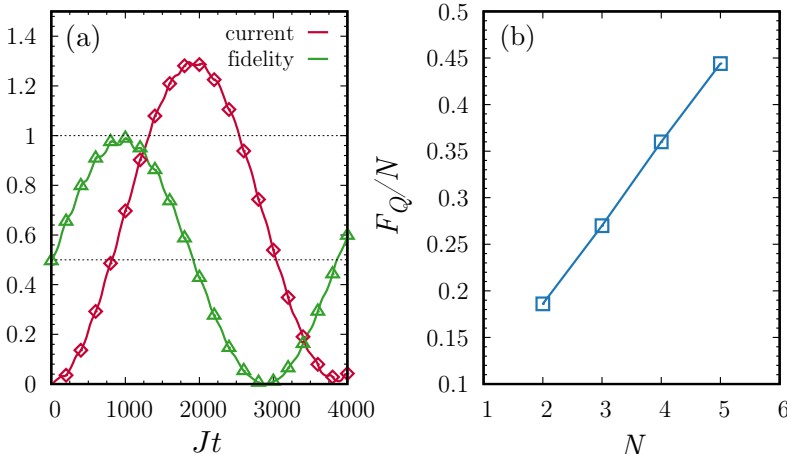

Figure 5: Creation of entangled states of angular momentum with quantum solitons. **a** Exact many-body dynamics of the current (in units of the hopping constant $J$) following a quench from $\Omega/\Omega_0 = 0$ to $\Omega/\Omega_0 = 1/2$. Here we set $N_s = 28$, $N = 3$, $U/J = -0.51$ and $\Delta_0/J = 0.015$. At one quarter of oscillation period, the superposition $|\psi\rangle = \frac{1}{\sqrt{2}}(|l_z = 0\rangle + |l_z = N\rangle)$ is formed, with a fidelity very close to 1. No fine tuning of parameter is required. **b** Scaling of the quantum Fisher information with particle numbers, showing that it reaches the Heisenberg limit $F_Q \propto N^2$. The system parameters are described in the supplementary material.

Remarkably, this procedure *dynamically entangles* the angular momentum state at $\Omega = 0$, i.e. $l_z = 0$, with the one at $\Omega = \Omega_0$, i.e. $l_z = N$ (see again Fig. 2), yielding $|\psi\rangle_{NOON} = \frac{1}{\sqrt{2}}(|l_z = 0\rangle + |l_z = N\rangle)$ when the current reaches the half of its maximum value. We note that such entangled states are superposition of current states, which are dual to the "NOON" states defined in the particle-number Fock basis [29].

The response of such a state to an external rotation is $|\psi(\phi)\rangle = e^{i\phi\hat{L}_z/\hbar}|\psi\rangle_{NOON}$, and the quantum Fisher information [30, 31] $F_Q = 4(\langle\psi'(\phi)|\psi'(\phi)\rangle - |\langle\psi'(\phi)|\psi(\phi)\rangle|^2)$, being $|\psi'(\phi)\rangle = \partial|\psi(\phi)\rangle/\partial\phi$. For our state we find $F_Q \sim N^2$, i.e. it reaches the Heisenberg limit - see Fig. 5. The corresponding sensitivity $\delta\phi$, therefore, is

$$\delta\phi \geq \frac{1}{(F_Q)^{1/2}} = \frac{1}{N}. \tag{10}$$

This shows that *entangled states of quantum solitons with different angular momenta lead to a quantum advantage of the sensitivity.*

### 3.3.1 Quench parameters

The oscillation period of the current, that is inversely proportional to the gap induced by the barrier at the crossing between $|\Omega_1\rangle$ and $|\Omega_2\rangle$, can be modified by tuning the potential barrier strength $\Delta_0$. The physical parameters we used to obtain the data shown in Fig. 5, panel (b) are obtained as follows: for each value of $N$ we choose $U$ in order to have the same spatial size of the many-body bound state as obtained by the study of the density-density spatial correlation function. The final choices are summarized in the Table 1 below.

Table 1: Choice of parameters for the study of the quench dynamics. For a given particle number, we have chosen the interaction stregth and the barrier strength in such a way that the many-body bound state has the same size. All the calculations are performed with $N_s = 20$.

| $N$ | $U/J$ | $\Delta_0/J$ |
|---|---|---|
| 2 | -1.06 | 0.05 |
| 3 | -0.72 | 0.03 |
| 4 | -0.52 | 0.025 |
| 5 | -0.40 | 0.01 |

# 4  Conclusions

In this work we present the first stages for creating a solitonic-based matter wave interferometer [32] and how one can harness many-body effects to obtain a precision beyond the standard classical limits [33–35]. A more thorough analysis for experimentally realistic scenarios can be explored including quantum/thermal noise [36, 37].

Summarizing, we have demonstrated that attracting bosons on a ring display fractionalization of angular momentum. On the fundamental level, such feature represents a remarkable extension of well known predictions for repulsively interacting bosons due to Byers-Yang-Onsager-Leggett [15, 19, 20]: The many-body bound-state nature of the ground state of attractive bosons implies fractional angular momenta per particle; interactions do not change the fractionalization on a continuous ring but they do affect it in the generic (lattice) system in which also the relative coordinate of the particles are sensitive to $\Omega$. These dependence is related to the formation of solitons/bound-states which can only be fully formed when their characteristic length is smaller than the system size, Such features are due to the entanglement in the ground state: the effect vanishes in the Gross-Pitaevskii limit in which the many-body wave function describes a factorized state. The $1/N$ fractionalization can be observed experimentally by studying the system's momentum distribution; the observation of such effect would provide the evidence of the formation of many-body quantum solitons beyond the Gross-Pitaevskii mean-field regime.

We note that, because of the formation of quantum solitons, an enhanced control on $N$ in the experiments is expected; in the lattice such value is protected by a finite gap [14]. The fractionalization of the angular momentum can define protocols to measure the number of particles in cold atoms experiments. Our results yield a $N$-factor enhancement in the sensitivity of attracting bosons to an artificial gauge field. We have provided a protocol to prepare a superposition of current states, explicitly exploiting the strong correlations, and we demonstrated that this state has a quantum Fisher information scaling as $N^2$, thus allowing to reach the Heisenberg limit in atomic interferometry.

# Acknowledgments

The Grenoble LANEF framework (ANR-10-LABX-51-01) is acknowledged for its support with mutualized infrastructure. We thank National Research Foundation Singapore and the Ministry of Education Singapore Academic Research Fund Tier 2 (Grant No. MOE2015-T2-1-101) and ANR SuperRing (ANR-15-CE30-0012) for support. LPL is a member DIM SIRTEQ (Science et Ingénierie en Région Île-de-France pour les Technologies Quantiques).

# A  Separation of center-of-mass and relative coordinates

We detail here the coordinate transformation to center-of-mass and relative coordinates. We introduce the Jacobi coordinates

$$
\begin{pmatrix} y_1 \\ y_2 \\ \dots \\ y_N \end{pmatrix} = M_{\text{Jac}} \cdot \begin{pmatrix} x_1 \\ x_2 \\ \dots \\ x_N \end{pmatrix} ,
$$

with the Jacobi matrix given by

$$
M_{\text{Jac}} = \begin{pmatrix}
1 & -1 & 0 & 0 & \cdots & 0 \\
\frac{1}{2} & \frac{1}{2} & -1 & 0 & \cdots & 0 \\
\frac{1}{3} & \frac{1}{3} & \frac{1}{3} & -1 & \cdots & 0 \\
\cdots & \cdots & \cdots & \cdots & \cdots & 0 \\
\frac{1}{N-1} & \frac{1}{N-1} & \frac{1}{N-1} & \frac{1}{N-1} & \cdots & -1 \\
\frac{1}{N} & \frac{1}{N} & \frac{1}{N} & \frac{1}{N} & \cdots & \frac{1}{N}
\end{pmatrix} . \tag{11}
$$

Here, $y_N = X_{\text{CM}} \equiv \frac{\sum_{l=1}^{N} x_l}{N}$ is the center-of-mass coordinate we want to separate out. The Jacobi matrix (11) is however not orthogonal (i.e. it is not a rotation). Nonetheless, the matrix $M_{\text{Jac}}$ can be easily converted to a pure rotation $R_{\text{Jac}}$ via the rescaling:

$$
R_{\text{Jac}} = \text{diag}\left( \sqrt{\frac{1}{2}}, \sqrt{\frac{2}{3}}, \dots, \sqrt{\frac{N-1}{N}}, \sqrt{N} \right) \cdot M_{\text{Jac}} , \tag{12}
$$

where diag(...) is a diagonal matrix, with the numbers in the parenthesis specifying the diagonal matrix elements. Indeed one can straightforwardly verify that $R_{\text{Jac}} \cdot R_{\text{Jac}}^{\top} = 1$, where $R_{\text{Jac}}^{\top}$ is a transpose of $R_{\text{Jac}}$.

We define then the coordinates

$$
\begin{pmatrix} z_1 \\ z_2 \\ \dots \\ z_N \end{pmatrix} = R_{\text{Jac}} \cdot \begin{pmatrix} x_1 \\ x_2 \\ \dots \\ x_N \end{pmatrix} .
$$

Note that $z_N = \sqrt{N} X_{\text{CM}}$.

Let us now introduce one final transformation, which brings us back to $X_{\text{CM}}$ as one of the variables, while keeping the Jacobian determinant of the transformation equal to one:

$$
\xi_l = N^{\frac{1}{2(N-1)}} z_l, \quad l = 1, 2, \dots, N-1,
$$

$$
\xi_N = \frac{1}{\sqrt{N}} z_N = X_{\text{CM}} .
$$

This defines the relative and center-of-mass coordinates used in the main text.

By a similar procedure one can identify the transformation to the Jacobi momenta $Q_l$, canonically conjugate to $\xi_l$, where $Q_N = \sum_{j=1}^{N} p_j = P_{\text{CM}}$ is the center-of-mass momentum. In particular, by introducing a set of momenta $\vec{P}_z = R_{\text{Jac}} \vec{p}$, with the same Jacobi matrix $R_{\text{Jac}}$ as the one used for spatial coordinates, one can show that $Q_l = \alpha P_{z_l}$ for $l = 1, \dots N-1$ with $\alpha = N^{-1/[2(N-1)]}$, and $Q_N = \sqrt{N} P_{z_N} = P_{\text{CM}}$.

The final Hamiltonian then reads

$$
\begin{aligned}
H &= \sum_{j=1}^{N-1} \frac{1}{2\mu_N} Q_j^2 + V_{int}(\xi_1, ..., \xi_{N-1}) \\
&+ \frac{1}{2M} (P_{CM} - Nm\Omega R)^2,
\end{aligned}
\tag{13}
$$

where $\mu_N \equiv N^{-\frac{1}{(N-1)}} m$ is the mass of the relative problem, $M = Nm$ is the total mass.

## B  Exact Bethe Ansatz results for the continuous ring

We start from the Lieb-Liniger model Eq.(1) of the main text, where we drop the constant $E_\Omega$:

$$
H_{LL} = \sum_{j=1}^{N} \frac{1}{2m} \left( p_j - m\Omega R \right)^2 + g \sum_{j<l} \delta(x_j - x_l).
\tag{14}
$$

For the Lieb-Liniger model, the total momentum and energy are $P_{CM} = \hbar \sum_{j=1}^{N} k_j$ and $E = (\hbar^2/2m) \sum_{j=1}^{N} k_j^2$ respectively, where the $k_j$ are obtained by solving the Bethe equations [22]

$$
k_j = \frac{2I_j \pi}{L} + 2\pi \frac{\Omega}{\Omega_0 L} - \sum_{\ell=1}^{N} \arctan\left( \frac{k_j - k_\ell}{c} \right),
\tag{15}
$$
$$
j = 1, ..., N,
$$

where $c = 2mg/\hbar^2$, $L = 2\pi R$ is the ring circumference and $I_j$ is a set of integer (semi-integer) numbers defining the state of the system. For repulsive interactions, all the $k_j$'s are real. For $2l\pi/L \leq \Omega \leq 2(l+1)\pi/L$, the ground states can be obtained by $I_j = -(N-1)/2 + j + \ell$, with integer $\ell$, yielding a center of mass momentum given by $P_{CM} = \hbar \sum_j k_j = \ell N \hbar/R$, as readily follows by noticing that $\arctan[(k_j - k_\ell)/c]$ is an odd function.

For repulsive interactions, the allowed values for the center of mass are integer multiples of $2p_F$, where $p_F = \hbar N/2R$, yielding

$$
E_{GS} = \frac{N\hbar^2}{2mR^2} (\ell - \Omega/\Omega_0)^2 + E_{int},
\tag{16}
$$

with $\Omega_0 = \hbar/mR^2$. The ground state energy hence results periodic in $\Omega$ with period $\Omega_0$ and the persistent current, obtained as $I_p = -(\Omega_0/\hbar)\partial E_{GS}/\partial \Omega$, reflects the center-of-mass quantization for any value of interaction strengths.

For attractive interactions the Bethe equations of Eq. (15) admits complex solutions and the ground state corresponds to a many-body bound state: $k_j = \kappa - i(n+1-2j)g/2$, $j = 1 \ldots n$. Such $n$ string solutions holds also for $\Omega \neq 0$, since the scattering matrix is not affected by $\Omega$. The ground state of Eq. (14) is made of a single $n = N$-string, yielding Eq.(3) of the main text. Here we point out that string hypothesis holds for $cL \to \infty$. The finite size corrections to the string solutions (for recent references, see [38, 39]) can affect the interaction energy $E_{int}$.

## C  Numerical Methods

Here we present the numerical techniques that have been used to obtain the results presented in this paper. We solve the eigenvalue problem by writing the Hamiltonian, $\hat{\mathcal{H}}$, as a matrix $H_{ij}$

in the Fock basis. This basis is then hashed in a more efficient form [40] in order to write the Hamiltonian in a sparse way. In particular, our numerical code is written in Python and the sparse Hamiltonian is diagonalized using ARPACK within the SciPy library. We have performed simulations with $N_s = 11$ to $N_s = 24$ sites and $N = 2$ to $N = 6$ particles, with a Hilbert space dimension up to $5 \times 10^5$, for different values of the flux $\Omega/\Omega_0$. Simulations have also been benchmarked with DMRG [41] data. After solving the eigenvalue problem, the correlation function $C_{lk} = \langle a_l^\dagger a_k \rangle$ is calculated using the ground state of the system and is used to obtain the time-of-flight results of Fig. 3.

## D  Two-particle exact solution

In the $N = 2$ sector, the Bose-Hubbard model the many-body wavefunction can be obtained using the coordinated Bethe Ansatz approach. Therefore, the ground-state energy and correlation functions can be accessed exactly. We generalize Ref. [42] to include the presence of an artificial gauge field in the Hamiltonian. Here, we gauge away the Peierls factors in the Hamiltonian and we impose twisted boundary conditions: $\hat{a}_{N_s+1} = e^{2\pi i \Omega/\Omega_0} \hat{a}_1$. A general two particle state can be written as:

$$|\phi\rangle = \sum_{j,k=1}^{N_s} \phi_{jk} \hat{a}_j^\dagger \hat{a}_k^\dagger |0\rangle, \tag{17}$$

where $\phi_{jk}$ is the two-particle wavefunction, symmetric under the exchange of $j$ and $k$, and normalized to unity. The energy of the system is found by solving the time-independent Schrödinger equation $\hat{H}|\phi\rangle = E|\phi\rangle$ using the Bethe Ansatz technique. In the center-of-mass and relative discrete dimensionless coordinates $X = (j+k)/2$, $x = j-k$ and $P = p_1 + p_2$, $p = (p_1 - p_2)/2$ the wavefunction $\phi_{jk}$ reads:

$$\phi_{jk} = e^{iPX} \left( a_{12} e^{ip|x|} + a_{21} e^{-ip|x|} \right). \tag{18}$$

The energy eigenvalues of the two-particle system are given by $E = -4J \cos(\frac{P+\Omega}{2}) \cos(p)$. The center of mass momentum is obtained by imposing twisted boundary conditions and quantization of the ring:

$$P_n = \frac{2\pi}{N_s}(n - 2\Omega/\Omega_0). \tag{19}$$

For the BHM the relative momentum $p$ is obtained by the condition:

$$(-1)^n e^{ip(N_s+1)} = y(P_n, p), \tag{20}$$

with

$$y(P_n, p) \equiv \frac{a_{21}}{a_{12}} = -\frac{\frac{U}{4J_0} - i \cos\left(\frac{P}{2}\right) \sin(p)}{\frac{U}{4J_0} + i \cos\left(\frac{P}{2}\right) \sin(p)}. \tag{21}$$

It is interesting to compare the BH and the Lieb-Liniger pictures. In the latter case, the equations to solve are

$$e^{ipL} = Y(p), \tag{22}$$

with

$$Y(p) \equiv \frac{a_{21}}{a_{12}} = -\frac{c - ip}{c + ip}. \tag{23}$$

Note that, in contrast with the BH case, Eqs.(22), (23) are decoupled, i.e. the center of mass momentum $P$ decouples to the relative momentum. As a result, the imaginary part of the momentum $p$ is independent on $\Omega$; this feature implies that the periodicity of the ground state energy does not change with the interaction strength. For the BHM, instead, $P$ and $p$ are coupled; this feature has a clear effect in the periodicty of the ground state energy. In conclusion, in the BHM the dependence of the periodicity on interactions is an effect of the coupling between center of mass and relative momentum.

Note that by solving Eqs.(20), (21) becomes fully determined. Thus, the time of flight images can be then readily evaluated by:

$$
\begin{aligned}
n(\mathbf{k}) &= \sum_{j,l=1}^{N_s} e^{i\mathbf{k}\cdot(\mathbf{x}_j-\mathbf{x}_l)}\langle a_j^\dagger a_l\rangle \\
&= \sum_{j,l=1}^{N_s} e^{i\mathbf{k}\cdot(\mathbf{x}_j-\mathbf{x}_l)+i\Omega(j-l)/\Omega_0}\sum_n \phi_{jn}^*\phi_{nl}.
\end{aligned}
\tag{24}
$$

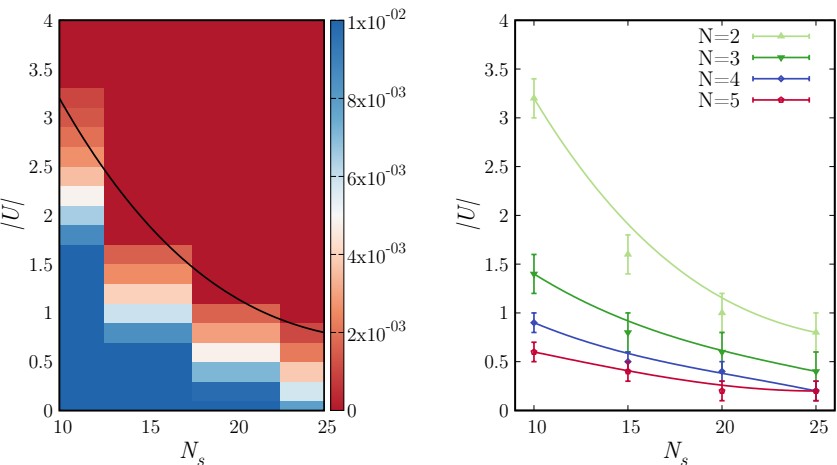

Figure 6: **a** Density plot of the renormalized energy difference between the $N$-times periodicity and the nonrotating system for $N = 2$. Solid lines gives the threshold for which $\mathcal{E}(U, N_s, N) < 10^{-3}$. In figure **b** we show the threshold given by condition $\mathcal{E}(U, N_s, N) < 10^{-3}$ for different number of particles and system sizes.

# E   Finite-size effects

In order to relate the size of the many-body bound state and the periodicity of the currents we analyze the dependence of the ground-state energy on the artificial gauge flux $\Omega/\Omega_0$ for various values of interaction strength $U$ and different system sizes $N_s$.

We estimate the spatial size associated to the many-body bound state by studying the exponential decay of the density-density correlations [14]

$$
\langle n_j n_{j+r}\rangle \approx \exp[-r/\xi].
\tag{25}
$$

We quantify the quality of the $1/N$ periodicity of the ground-state energy $E(\Omega)$ by calculating

$$
\mathcal{E}(U, N_s, N) = \frac{|E(\Omega = \Omega_0/N) - E(\Omega = 0)|}{E(\Omega = 0)},
\tag{26}
$$

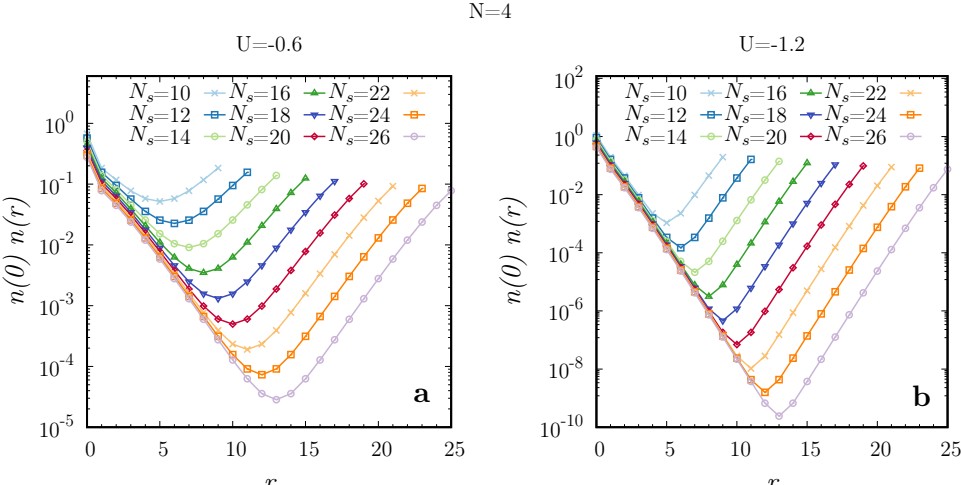

Figure 7: Density-density correlations $C_{j,j+r}$ for $N = 4$ within and outside the regime where the system presents an increase of the periodicity of the current.

such that $\mathcal{E}(U, N_s, N) = 0$ corresponds to a perfect $1/N$ periodicity.

Figure (6)(a) shows the density plot $\mathcal{E}(U, N_s, N)$ for a fixed number of particles $N = 2$. In this figure, we show that for large $U$ and a sufficiently large system size, the periodicity of the ground-state energy is increased by a factor $N$ with respect to the noninteracting case. In Fig. 6)(b) we calculated the threshold for which the minimum of the $N$-time periodicity is obtained within an error of 0.1%, i.e. $\mathcal{E} < 10^{-3}$, for different number of particles (corresponding to the solid line in Fig. 6)(a)). Finally we compare the density-density correlations $\langle n_j n_k \rangle$ for two different points in the density plot shown in (a), one within the region where the current presents $N$-time periodicity and one above the threshold. Indeed, comparing Fig.7 and Fig.8, we can demonstrate that the size of the soliton, which depends on $U$ for a fixed number of particles, must be smaller than $N_s$ in order to observe the enhanced sensitivity presented in this paper.

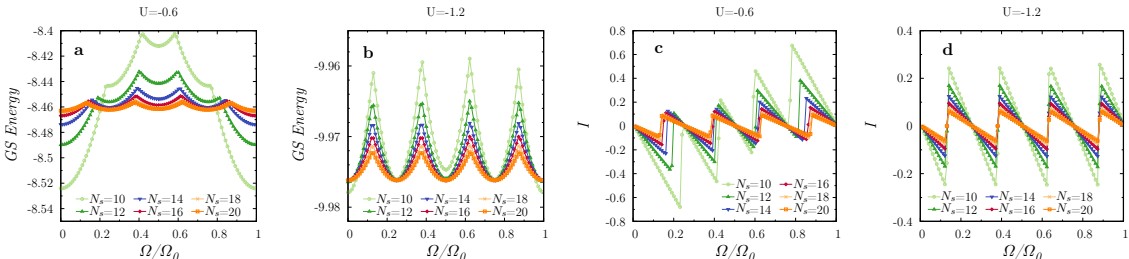

Figure 8: **a**, **b** ground state energy for different interaction as a function of the flux. **c**, **d** current, calculated as $I = -\partial E_{GS}(\Omega)/\partial\Omega$, for different interaction as a function of the flux.

## F  Mean-field theory

Here we provide extra data on the mean field Gross-Pitaevskii, which in the rotating frame is given by:

$$i\hbar\frac{\partial}{\partial t}\Psi = \left[\frac{\hbar^2}{2m}\left(-i\frac{\partial}{\partial x} - \frac{2\pi}{L}\Omega\right)^2 + g|\Psi|^2\right]\Psi. \tag{27}$$

For repulsive interactions, mean-field theory predicts no change in the energy landscape, which is made of parabolas all centered around integer values of the flux quantization $\Omega_0$. However, for the attractive case, where solitons are formed [43], the circulation is no longer quantized, and the system reacts to the induced rotation. This breaking of the quantization reflects on the current as a smoothening with no discontinuity at half values of the quantized circulation. Figure 9 shows how this transition occurs for different values of the interactions, both in the rotating frame and in the lab frame.

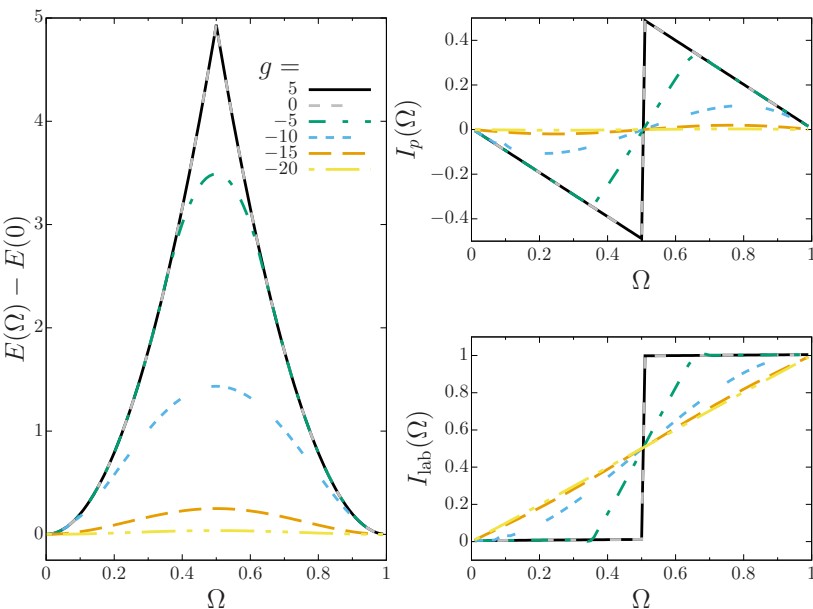

Figure 9: Mean-field simulations of the Gross-Pitaevskii equation, Eq. (27), for both attractive and repulsive interactions. The left panel shows the energy landscape as a function of the flux $\Omega$, while the right panels show the current and current in the lab frame from top to bottom, respectively.

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
