# Peer review of "Enhancing sensitivity to rotations with quantum solitonic currents"

_SciPost Physics, doi:SciPost Phys. 12, 138 (2022)_

## Round 1 · Referee Report · Anonymous (Referee 1) · 2020-12-1

Report

P. Naldesi et al. analyze in their preprint "Enhancing sensitivity to rotations with quantum solitonic currents" the ground state properties of a one dimensional Bose gas with strong attractive interactions in a ring geometry in presence of an artificial gauge field. The latter is induced by a global rotation with a given angular frequency. The authors mention as possible platform ultra-cold atoms and in particular atomtronics.

In the beginning P. Naldesi et al. analyze a modified Lieb-Liniger Hamiltonian where all atoms experience a rotation with frequency Omega. Using similar techniques as in Ref.[31] [Phys. Rev. 130, 1605 (1963)] they can calculate the ground state energy analytically, Eq.(2), and predict the effect of angular momentum fractionalisation. The latter describes the effect that the total angular momentum increases in integer steps from 0 to N (number of atoms) when the frequency Omega is increased from 0 to a period Omega0 that is determined by the mass of the atoms and the radius of the ring. Consequently, the angular momentum per particles increases in fractional steps 1/N.

For the remainder of their manuscript, the authos analyze a Bose-Hubbard model that originates from the Lieb-Liniger Hamiltonian but with an additional lattice with Ns sites. This Hamiltonian describes an attractive onsite interaction and a nearest-neighbour hopping with a phase that is controlled by the angular frequency Omega. They use exact diagonalization to find the ground state and verify that this model also shows angular momentum fractionalisation. The original picture of the 1/N-periodicity is reproduced for sufficiently strong attractive interactions. Analyzing the momentum distribution, the authors find fractional steps in the mean-square radius. Thus the fractionalisation can be in principle measured via time-of-flight imaging. Furthermore, they cannot reproduce this result with a Gross-Pitaevskii equation why they claim that these effects are purely quantum many-body effects.

In the end, P. Naldesi et al. provide a technique to use this system for potential metrological applications in sensing rotations. In order to show this they interrupt the ring by a localized barrier at a certain site and quench the angular momentum from Omega=0 to Omega=Omega0/2 (half of a period). With this they show that the system, after a certain time, reaches a coherent superposition of two total angular momentum states corresponding to a total angular momentum of 0 and N, respectively. With this state they show that the system can in principle reach the Heisenberg limit where the variance of the estimated phase scales as 1/N^2.

Comment:

The physics that the authors describe is interesting, timely, and new. In my opinion, especially the analysis of the angular momentum fractionalisation is well explained and the possibility to measure it in state-of-the-art atomtronics experiments via time-of-flight is remarkable. The "enhancement of sensitivity to rotations" part of the preprint is less clear since the protocol for the dynamical entanglement seems to me kind of arbitrary. For instance it is not really clear if that protocol works independent of N, Delta_0, Omega_0, U/J, and N_s.

I believe that this paper meets two Expectations at least partially that are required to be accepted for SciPost Physics:

Expectation 1; Detail a groundbreaking theoretical/experimental/computational discovery:
I am not sure if the findings are groundbreaking but they are certainly very interesting and might be measurable in current experiments.

Expectation 4; Provide a novel and synergetic link between different research areas:
This work shows a nice link between atomtronics and the many-body effects of strongly interacting matter in presence of an artificial magnetic field. The novelty, in my eyes, is the physical effect and the possibility to realize it.

In my opinion this manuscript does not yet meet all general acceptance criteria. The two criteria that I have in mind are:

Criteria 3; Provide sufficient details:
I think it is hard to reproduce the results and it would be a good idea to clarify a few things and maybe add to the appendices. I will clarify this in "Questions".

Criteria 5; Provide all reproducibility-enabling resources:
Same as above.

In general, I think, this manuscript should be considered for publication in SciPost Physics. But before supporting publication unequivocally I would like that the authors address the following questions and criticism.

Questions:

(i) In general it should be clarified what exactly is meant by "a nontrivial generalization of a theorem due to Leggett". Is the reason for claiming this that the period for the energy is a fraction of the original period?

(ii) Regarding the previous question: What are the requirement for this fractionalisation of the period? Even in the Bose-Hubbard model it seems to be only true in the large |U/J| limit. What is the role of attractive interactions in such a generalized Leggett theorem? Because it seems that it depends at least on the sign of the interaction strength.

(iii) Already in Eq. (1), I am a little confused what pj and xj are. What I understood is that xj is proportional to an angle that marks the location of the atom on the ring. And pj is its conjugate that is proportional to the derivative with respect to that angle. Is that correct?

(iv) I believe that Eq. (B2) is essentially a generalization of Eq. (2.15) in Phys. Rev. 130, 1605 (1963). I would appreciate if the authors could add details how this equation is derived. Also, the authors should add more details how they solve this equation for attractive interactions. The solution for kj is not even a proper equation in Appendix B and it is unclear (at least to me) how they derived this equation. I think this is crucial to understand the properties of the bound state.

(v) The authors introduce the Bose-Hubbard model without even mentioning how it is derived. I guess this is the typical tight-binding and lowest band approximation but I am more curious how the lattice and the corresponding Wannier functions would look like. I imagine the authors consider an optical lattice that is produced by interference of a Laguerre-Gauss mode with a plane wave a la Phys. Rev. Lett. 95, 063201 (2005). The authors assume that the Wannier functions are Gaussian functions with a given width. How does that fit to the ring geometry and the fact that there is a radial and an "angular" confinement. This also leads to my confusion regarding question (iii). Maybe it would be a good idea to mention how Eq. (3) is derived, e.g. in an additional appendix.

(vi) After Eq. (3) the authors write J,U<0 but later U/J<0 (e.g. Fig.2).

(vii) The authors should write how many sites they have used in the caption of Fig.2. Also, the authors mention that the results agree with the analytical result in the appendix. Is that visible in that figure or has that just been verified elsewhere?

(viii) Regarding the 1/N period in the Bose-Hubbard model: the authors write that this only happens for sufficiently large |U/J|. It would be useful to quantify this. Also, is the period exactly 1/N for finite but large |U/J|?

(ix) I think that the comparison with the Gross-Pitaevskii equation is interesting but the analysis does not convince me. I would expect that the Gross-Pitaevskii can only potentially explain the correct behavior for N=1 and N to infinity. For very large N the stairs of the staircase in the exact result become shorter and shorter, so could it not be possible that the angular momentum and the energy converge in the N to infinity limit to the GPE result?

(x) Should the x axis in Fig 3 be Omega/Omega0?

(xi) Regarding the dynamical entanglement: does this protocol work in general? How does it depend on the system parameters? I ask this question because the results are shown only for a very specific choice. How large could N be? Is that independent of Ns, so could one in principle also work in the regime N>Ns?

(xii) The Lz=0 and Lz=N are states that are not fractionalised if I understand it correctly. Is the effect of angular momentum fractionalisation required for producing this state? Your parameter U/J=-0.51 does not seem to be in the very strong interaction limit.

(xiii) After one quarter of the oscillation period the fidelity approaches 1, so the state seems to be the "NOON" state. For this state the authors calculate that FQ=N^2. From this derivation I would expect that the quantity FQ/N in Fig.4(b) should be N. Why is that not the case? Also, the authors should add (a) and (b) in Fig.4. The x axis in (a) should be J*t I guess?

(xiv) In appendix D and appendix E the authors write L in the different figures but I think this should be Ns.

(xv) In appendix C the authors mention what are the system sizes they have analyzed. Is the dimension of the Hilbert space (N+Ns-1) choose N? Does the Hilbert space dimension of up to 10^6 correspond to the case 29 choose 6 = 475020. If this is true, I would appreciate if the authors could give the formula such that it is clear where the 10^6 comes from.
  • validity: -
  • significance: high
  • originality: high
  • clarity: good
  • formatting: good
  • grammar: -

Author:  Piero Naldesi  on 2021-11-25  [id 1971]

(in reply to Report 1 on 2020-12-01)

We thank the Referee I for his/her positive recommendation and for the careful reading of our paper.

We addressed all the posed questions, added details in the main text and in the appendices.

(i,ii,viii) Leggett's theorem [A. Leggett, inC.W.J. Beenakker, et al, Granular Nano-electronics (Plenum Press, New York, 1991) p. 359.] states that the periodicity of the persistent currents is fixed by the flux quantum of the system. As an example in a superconductor, the doubling of the quantum of flux due to the formation of pairs results in a $1/2$ of the periodicity of the persistent current. The formation of quantum solitons composed of $N$ particles in our system magnifies the quantum of flux by a factor of $N$. This effect translates into a reduction of the period by $1/N$.
The fractionalization, and therefore the reduction of the periodicity, is a result valid for every $U/J<0$ provided that the size of the system is large enough to allow the soliton to form. As for every bounded object, solitons are confined over a characteristic length that increases when $|U/J|$ decreases. If this length is smaller than the system size then the soliton is properly formed. More details on the quantification of the threshold $U/J$ for having a reduction by $1/N$ of the periodicity, can be found in appendix E (in particular in fig 5).
We modified the main text to make these points clearer.

(iii and iv) $x_j$ is the position coordinate of the $j-th$ particle over the ring and $p_j$ is its conjugate variable. We have now added their definition in the main text.

Equation B2 is a set of implicit equations, known as Bethe Ansatz equations. The effect of the flux was established in many works, see for instance [, Sutherland, Shastry twisted boundary conditions]. Solving this system of transcendental equations give us the momenta of all the particles from which we can calculate the energy of the state. We added the reference mentioned by the referee in the appendix.

(v)
The Bose Hubbard model [
\textit{J. Hubbard}, Proc. Roy. Soc. London A 276, 238 (1963); \textit{F. D. M. Haldane}, PRA 80.4 (1980); \textit{M.P.A Fisher}, PRB 40, 546 (1989) ] has became nowadays a milestone between the quantum many body system experimentally achievable, especially after some years ago it has been shown how to realize it in a ultracold atom set up \textit{D. Jaksch, P. Zoller} Annals of Physics, Volume 315, Issue 1, January 2005, Pages 52-79].
For more information on how experimentalist do it see for example [\textit{Marc Cheneau} Nature volume 481, pages484–487 (2012), \textit{Choi} et al. Science 24 Jun 2016: Vol. 352, Issue 6293, pp. 1547-1552, \textit{Rubio-Abadal} et al. Phys. Rev. X 10, 021044 – Published 27 May 2020]. We included the corresponding reference in the main text.

(vi) We have changed the sentence for better clarity.

(vii)
We added numerical details in figure 2.
The steps obtained from the numerical analysis are exactly those predicted in equation (2).
Indeed, equation (2) predicts a perfect $1/N$ fractionalization of the angular momentum, which is accounted by the $N\frac{\Omega}{\Omega_0}$ term of the equation. We have clarified this point by explicitly mentioning the relation between the average current per aprticle $\mathcal{I}_p$ and the quantum number $\ell$ in the main text.
In appendix we show how to obtain equation (2). The analytical calculations, are based on the string hypothesis. In its regime of validity, namely $cL\rightarrow\infty$, they agree perfectly with the numerical results.

(ix) While for $N=1$ GPE clearly reproduces the single particle case. In the limit $N\rightarrow \infty$ the equivalent condition is more involved. In fact one can not just take the limit $N\rightarrow \infty$ but needs to keep the energy of the state finite by sending $g\rightarrow 0$. For a precise derivation of this condition see \textit{L. Piroli and P. Calabrese}, Phys. Rev. A 94, 053620 (2016). ].
To address this point, we have written an additional appendix where we include more data and discussion on the Gross-Pitaevskii results.

(x) We changed the label of Fig 3.

(xi) Indeed if we want to create an entangled state made out of solitons with different currents, we first need to create fully formed solitons and then to entangle them.
The answer to this point can therefore be divided in two different points: (a) for which parameters we have a well defined quantum soliton in the ground state of our system and (b) for which parameters the protocol works.

(a) The condition for the presence of well formed solitons in the system is a combination of $N$, $N_S$ and $U$ (see also answer to point (i,ii,viii) ).
Given $N_s$ and $N$, one should take an interaction strength $U$ small enough to make the size of the soliton $\xi$, in unit of the lattice spacing, smaller than the size of the system $\xi < N_s$. Since the bounding energy of the soliton is proportional to $|U| N^2$, to keep the size of the soliton fixed one should play inversely with the two parameters $U$ and $N$, if one grows the other must decrees. For more details on the soliton size $\xi$ see also [P. Naldesi PRL 122, 053001 (2019)]. The system for $N>>N_s$ has been also studied in a different regime (repulsive) by part of the authors [E. Compagno, Phys. Rev. Research 2, 043118, 2020]. Concluding, yes, if we fix the number of lattice sites to $N_s$, one in principle could make the number of particles grow larger than $N_s$ and setting the interactions in order to verify the condition $\xi < N_s$.

Answering to the second part, (b), the protocol we proposed is pretty general, and it allows to entangle the two target ground states of the system at $\Omega_0$ to $\Omega_1$ by quenching at the value of flux, $\Omega_f$, for which these two states are degenerate in energy. However, an inhomogeneity must also be introduced in the system which, in our case, consists of a delta-like barrier. The first reason for which the inhomogeneity is necessary is to split the degeneracy point and create a superposition, symmetric and antisymmetric, at the crossing point. To do so the strength of the barrier must create an approximate 2-level system, without coupling other states in the spectrum. The second reason to add a barrier is to break the symmetry of system, since a system invariant under discrete rotations is insensitive to a change of the flux.

(xii) The threshold for having a fully formed soliton is quite involved, as it depends on $N_s$, $N$ and $U/J$ (see previous answers and Fig.5 of the appendix), therefore $U/J=-0.51$ is strong enough for the particular case studied in the quench protocol.

Fractionalization is a phenomenon that occurs at every flux, provided that interactions are strong enough (see discussion above).

In particular, in the ground state of the Hamiltonian, for a general $\Omega$, the angular momentum would be $L_z=l_z/N$. Thus, for the two states we are considering have $l_z=0$ and $l_z=N$. In other words, with respect to the inset of Fig.~2(a), to produce the NOON state, we need to connect the first parabola with the $N$th one (quantum number $\ell \in \{0\rightarrow N\}$).

(xiii) We changed label and added all the details in fig 4.
The Cramer-Rao bound states that the Quantum Fisher information scales like $N^2$ for "NOON" states, still there can be a proportional coefficient in front. Indeed in our case $FQ/N$ scales like N. as shown in Fig 4.

(xiv) We uniformed the notation.

(xv) The dimension of the Hilbert space is indeed $\binom{N+Ns-1}{N}$. We correct the estimanation of the size of the largest Hilbert space we studied to $5 * 10^5$.

---

## Round 1 · Referee Report · Anonymous (Referee 2) · 2020-12-12

Strengths

1- The proposed effect is of general interest and new.

2-The manuscript is of interest to a broad audience, and should also be of interest to a sizable community.

Weaknesses

1- There is no real discussion on the feasibility/usefulness of the measurement scheme. Important questions remain undiscussed.

2- The presentation can be improved (in particular the introduction is very unspecific).

3- It's disappointing that there are no larger-system calculations, and no calculations with imperfections are presented.

Report

The manuscript theoretically discusses a setup with attractive bosons trapped in a 1D ring lattice, in presence of an artificial gauge field, e.g. induced by a global rotation of the ring. It is analyzed how this leads to steps in the angular momentum that is extracted from ground-state energies. This "fractionalization" depends on both the number of atoms and the interaction strengths. It is explained by "bound soliton states", and it is shown to be a true many-body quantum effect (beyond mean-field). The fractional steps are showing up in time-of-flight momentum distributions. While those bound states have been studied by a sub-set of the authors in detail previously, here as a new feature they also introduce a quench scheme, which allows to dynamically produce an entangled state, i.e. a superposition of angular momentum states. This state is argued to be useful for "quantum enhancing" sensitivity in rotation measurements.

The effect is generally interesting, and an experimental realization of the proposed physics seems plausible. The manuscript is written in an accessible form for a broader audience, and should also be of interest to a sizable community. I think this work should be published in SciPost Physics, in particular I think that it fulfills the criteria "3.", i.e. it can be a starting point for follow-up work on quantum enhanced sensing.

However, there are some points which are a unsatisfying, as described in the following.

Requested changes

1- The entanglement generation, and rotation measurement scheme is lacking a discussion of the usefulness of the scheme. The Heisenberg limit scaling is only demonstrated for N=2,3,4,5.

  • I guess for a meaningful quantum enhanced sensitivity, one would want to go to the large N limit. From the manuscript, I don't get an intuition about this limit. From the scaling of parameter choices with N (table I), it seems that |U| -> 0 and Delta_0 -> 0 in this limit. Where does this go for large N? There needs to be a discussion on this. It's a bit disappointing that only such small Ns are considered, given that with MPSs/DMRG one could easily access much larger systems, so why not large N?

  • Similarly, is there a general issue with time-scales? One notices that to reach the optimal state, already for N = 3, one needs 1000 tunneling times. In an optical lattice experiments, I would expect that effects such as spontaneous photon absorption would then become very relevant. Is there a scaling of this time-scale with N? What determines the oscillation frequencies, both the long ones and the small short ones that are visible? It is written in the appendix that it's related to Delta_0, but how? What limits the fidelity, why does it not reach one exactly? Is the scheme robust to imperfections? I don't think extra calculations are necessary, but those question should be discussed honestly.

2- General presentation:

  • With a single paragraph, the discussion of the quench scheme is generally too short (given that it's a main result). I suggest to at least move the whole appendix F to the main text. It is needed there, because things like Delta_0 are not defined in the main text, and the parameter choices for U and Delta_0 are really important for Fig. 4 .

  • This may be a matter of taste (and I don't feel too strongly about it), but the beginning of the abstract and the introduction is extremely vague. I think the paper is trying to draw a too big of a picture, basically only saying that true quantum technology relies on entanglement. I don't really see any connection of this work to quantum supremacy in google's quantum computing efforts, or any connection to quantum simulation. It would be much nicer to put more emphasize on the actual systems of interest for the proposed setup of the paper. Having more specific information e.g. for BEC on chip type experiments would make more sense. There is also no mentioning of schemes using squeezed states of BECs in ring traps, which already exist [e.g. PRA 93, 023616 (2016)]

Other points:

  • The observation of the fractionalization in time-of-flight measurements is nice, but it maybe hard to obtain enough sensitivity, given the small N needed to see a clear fractionalization. Probably, a direct observation e.g. of density-density correlations is more feasible in quantum gas microscopes? That could be mentioned.

  • On the second page it reads

"Before treating the general case of the lattice ring, we will first assume that the density N/L of bosons, ..., is small enough to describe the system through the continuous Bose-gas integrable theory or equivalently the Lieb-Liniger model."

This sounds strange. I guess what they want to say is that they first consider analytical solutions for a ring-trap with contact interactions and then go to a lattice with on-site interactions. I guess to connect the two regimes one not only needs the low density limit, but somehow also N_S >> 1?

  • I suggest to move the definition of how the angular momentum is computed from the caption of Fig. 2 to the main text. It's kind of important, and bizarre for a caption, because it's not clear what the other parameters are.

  • It is clear how the scheme would be used for enhanced rotation measurements. It seems to be loosely suggested, that one could also measure real fields (e.g. "Our results yield a N-factor enhancement in the sensitivity of attracting bosons to an external field.") How would the neutral particles couple to a real EM field?

Minor points and typos:

  • There are notation switches, e.g. sometimes L is used for N_S (Fig. 6/7)

  • There is also a mix of calling the appendix "supplemental material" or "method section".

  • Typo: "Heisenberg" is often misspelled as either "Heinsenberg" or "Heiseberg".

  • "For repulsive interactions, independently of the interaction, EGS results periodic ..." -> independent of "interaction strengths" and "is periodic"

  • "bosons dynamics" -> "boson dynamics"

  • Definition of the BHM is a bit imprecise. Could be worth to re-iterate "periodic boundary conditions" that are used. Also J, U < 0 reads like it would imply J < 0.

  • analize -> analyze

  • Fig. 5: The shaded plot looks very strange. I assume it implicitly uses some interpolation, which is bizarre since Ns is a discrete variable

  • strenght -> strength

  • quech -> quench

  • round state -> ground state

  • ie -> i.e.

  • validity: good
  • significance: high
  • originality: good
  • clarity: ok
  • formatting: acceptable
  • grammar: reasonable

Author:  Piero Naldesi  on 2021-11-25  [id 1972]

(in reply to Report 2 on 2020-12-12)

We thank Referee II for his/her positive recommendation and for the careful reading of our paper.

We reshaped part of the introduction and the section on the quench scheme. We addressed all the posed questions, add details in the main text and in the appendices.

1) We fully agree that to prove the Heisenberg limit, one would want to go to the large N limit. Nevertheless, for this system, it is really difficult to access. We want to emphasize that with the use of any approximate method we directly lose the possibility of calculating the fidelity over a particular state. On top of that, the attractive interactions populate in every site all the possible multiple occupations, prohibiting to cut the local Hilbert space. We think that with state-of-the-art numerical techniques, by making use of an HPC cluster, one could push the number of particles a bit higher, however one could never enter the large N limit. In addition, such small changes would not modify the message of our publication.

Regarding the particular choice of the parameter, we fixed U in order to have the same size of the soliton for a different number of particles. Since the interaction energy is proportional to UN^2, by increasing any of these two parameters, U and N, we make the soliton shrink. Therefore to constrain the soliton in the same portion of space, if we increase N, U must decrease.

The discussion on the dynamical creation of the entangled state has been enriched in the main text.

The authors have studied in the detail the formation of a qubit-like state between different angular momentum states using the physical phenomena described in this work. Further details on the system-parameter dependencies can be found in [arXiv:2012.06269].

2) We improved the section on the quench scheme, adding details and moving there part of appendix F.

Following the referee's comment, we have significantly reduced the introduction. We have only kept parts that are directly related to our system, which focuses more on the introduction of the ideas and system we are investigating.

Unluckily fractionalization of angular momentum cannot be observed from density-density correlations (or any other power of the operator n_i) since in this operator the phase factor vanishes see eq (21) of Physical Review A 101, 043418 (2020). However, it's possible that implementing some more complex measure scheme in a quantum gas microscope can let us access this phenomenon but we think that such implementations go beyond the scope of the current work.

Indeed the low-density limit is reached by keeping the number of particles fixed and taking the limit N_S >> 1. For sake of clarity, we rephrase the sentence.

Following the suggestion, we moved the definition to the main text and expanded the discussion for better clarity.

We have rephrased the sentence. The referee is, of course, correct that neutral atoms cannot be directly coupled to E.M. fields; we have changed it to artificial gauge fields.

We introduced all the minor corrections in the text.

---

## Round 1 · Referee Report · Anonymous (Referee 3) · 2021-1-18

Strengths

The fractionalisation of angular momentum, its possible observation and the application to metrology are interesting observations.

Weaknesses

I am not sure the all the main results are new. The fractionalisation of angular momentum has probably been discussed elsewhere (by some of the authors). The possible observation by time of flight hardly fits the small-particle number limits discussed in the text. The application to metrology is poorly characterised.

Report

The manuscript “Enhancing sensitivity to rotations with quantum solitonic currents” discusses a boson particles in a 1D ring-shaped potential. The interesting result is that, in the case of interaction, the system is characterized by a fractionalization of angular momentum. The authors further discuss applications in metrology.

I think the paper is borderline for SciPost. On the one hand, the fractionalization result is certainly very interesting but probably not new (see [3-5]). The possibility to observe the fractionalization through time-of-flight experiments is also interesting and probably the really novel part here. However, there is no real deep discussion about the experimental relevance of this work. There are a number of strict conditions:” the 1D and ring geometry, the control of attractive interaction, etc. whose experimental relevance is not discussed. Furthermore, the number of particles in the system is really small, 4-to-6 atoms: is it possible to observe the time of flight of 6 atoms experimentally? I understand that increasing the particle number in the simulations may be numerically demanding/impossible.

The discussion about the application in metrology is really poor. I am probably fine with the calculation of the quantum Fisher information, but the discussion about the generation of the highly entangled state is too brief. This is one of the central point of the paper but the discussion, which covers only the two sentences “The ring is interrupted by a localized barrier … half of its maximum value” is certainly too vague. This should be expanded and detailed: what is the localisation barrier here? Is the generation of the entangled state robust against some source of noise? Figure 4(a) show that the starting point (t=0) has already a high fidelity with the NOON state: why? Probably there is no need to evolve the system to the NOON state: the state at t=0 is maybe already metrologically useful. To see this, I suggest to plot the Fisher information as a function of time.

Several details are put in the Appendix. First, the text never recalls the Appendix. Second, the separation between technical details and general discussion is too sharp, giving the general idea that the theory is discussed too briefly in the main text.

My overall feeling is that the physics is fine and interesting, but the paper is badly written. As it is, I would say that the paper suits better a more specialized journal. There are already many papers proposing neutral atoms for entanglement-enhanced metrology applications and several proposal to generate NOON states: the authors should make clearer what is the advantage of their system in terms of robustness to noise and/or possible (new) applications.

Further points:

1) The first part of the abstract “Quantum mechanics is characterized … entanglement properties.” Is too dispersive. There are many experiments already showing coherence and entanglement, I do not think the discussed system is particularly outstanding in this respect. So, I do not understand the relevance of the discussion in the first part of the abstract and of the introduction.

2) The author target metrology with their system. However, the paper is not put well in this context: - Atom (laser) gyroscope has been studied by M. O. Scully and J. P. Dowling, Phys. Rev. A 48, 3186 􏱗1993􏱐; Dowling PRA 57, 4736 (1999). See Cronin et al RMP 2009 for a review on atomic systems. - There are also several papers on metrology with neutral atoms, especially in the double-well potential, see Esteve et al Nature 2008, Berrada et al Nat. Comm. 2013, Trenkwalder et al Nat. Phys 2016. - Finally, there are also papers exploring the creation of NOON states with Bose-Einstein condensates. Please provide a (better) comparison between the results and ideas of this manuscript and the existing literature.

Requested changes

See report

  • validity: top
  • significance: good
  • originality: high
  • clarity: ok
  • formatting: good
  • grammar: excellent

Author:  Piero Naldesi  on 2021-11-25  [id 1970]

(in reply to Report 3 on 2021-01-18)

We thank Referee III for the careful reading of our paper.

We do not agree with referee III on the first part of this section.

Reference [3] studies a different Hamiltonian than the ones we studied. They looked at a quantized version of the GPE, in the large N limit. Nonetheless, their results point out towards the good direction, and they found a fragmentation that resembles our fractionalization. However, note that their results only affect the region around the usual level crossing and do not produce a 1/N fractionalization of the current as of the one we presented here.
References 3 and 5 do not include any form of artificial gauge field or rotation.
Therefore, we think that, to the best of our knowledge, our work presents a novel phenomenon not discussed in the literature before.

We fully agree that to prove the Heisenberg limit, one would want to go to the large N limit. Nevertheless, for this system, it is really difficult to access.
We want to emphasize that with the use of any approximate method we directly lose the possibility of calculating the fidelity over a particular state.
On top of that, the attractive interactions populate in every site all the possible multiple occupations, prohibiting to cut the local Hilbert space and thus, the use of other approaches such as DMRG or MPS. We agree that the conditions are strict, however, any real implementation of many-body quantum phenomena relaying on many-body states such as NOON states would have similar limitations. To the best of our knowledge, there is no experiment able to create NOON states made of current states (at the moment), and our work could potentially show such phenomena in the small system under controlled parameters.

a) Following the suggestion the paper has been reshaped and many details have been added. The section on the preparation of the NOON state is now wider and much more clear.
We want to point out that the state at t=0 is not at all "metrologically useful". This state has a large projection over the NOON state just because is "half" of it. Nevertheless, its QFI does not scale like N^2.

b) Following the referee's comment, which has been also pointed out by Referee 2, we have significantly reduced the introduction. In particular, we have put more focus on the system we are investigating and on the features that we have specifically looked at.

c) The main focus of our study is to prove that attractive bosons can lead to useful states in quantum metrology, framing this paper as a proof of principle.
In fact, here we did focus on the physical phenomenon in itself and such a thorough comparison is beyond the scope of the current work.
We completely agree with Referee III that comparing with other metrological devices is important and it is part of our current research.

---

## Round 2 · Author Response

We are grateful to all the referees for their careful reading and assessment of our paper.
We have addressed all the comments/criticisms made by the Referees in this response and in the revised version of the manuscript.
We hope that the paper can now be considered for publication.
Yours sincerely,
Piero Naldesi on behalf of the authors.

---

## Editorial Decision

published